# Identifying Outliers of the MODIS Leaf Area Index Data by Including Temporal Patterns in Post-Processing

**Baibing Ma** [1,2,3] and **Ming Xu** [1,2,3,*]

1. Synthesis Research Centre of Chinese Ecosystem Research Network, Key Laboratory of Ecosystem Network Observation and Modelling, Institute of Geographic Sciences and Natural Resources Research, Chinese Academy of Sciences, Beijing 100101, China; mabb.18b@igsnrr.ac.cn

2. College of Resources and Environment, University of Chinese Academy of Sciences, Beijing 100190, China

3. Jiangmen Laboratory of Carbon Science and Technology, Hong Kong University of Science and Technology, Jiangmen 529199, China

* Correspondence: xuming@hkustgz-jcl.ac.cn; Tel.: +86-13552593273

**Abstract:** The moderate resolution imaging spectroradiometer (MODIS) calculates the leaf area index (LAI) for each pixel without incorporating the temporal correlation information, leading to a higher sensitivity for the LAI that produces uncertainties in observed reflectance. As a result, an increased noise level is observed in the timeseries, making the data discontinuous and inconsistent in space and time. Therefore, it is important to identify and handle the outliers during the post-processing of MODIS data. This study proposed a method to identify the MODIS LAI outliers based on the analyses of temporal patterns, including the interannual and seasonal changes in the LAI. The analysis was carried out utilizing the data from 278 global MODIS LAI sites and the results were verified against the measurement obtained from 52 ground stations. The results from the analyses detected 50 and 92 outliers based on 1.5σ and 1.0σ standard deviations, respectively, of the difference between the MODIS LAI and ground measurements; correspondingly, 46 and 65 outliers, respectively, were identified by incorporating temporal patterns during the post-processing of the data. The validation results exhibited improved values of the coefficient of determination ($R^2$) after eliminating the MODIS LAI outliers identified through the interannual and seasonal patterns. Specifically, the $R^2$ between the ground measurement LAI and MODIS LAI increased from 0.51 to 0.54, 0.88, and 0.90 after eliminating MODIS LAI outliers when considering the interannual patterns, seasonal patterns, and both the interannual and seasonal patterns, respectively. The results from the study provided valuable information and theoretical support to improve MODIS LAI post-processing.

**Keywords:** MODIS LAI; outliers; temporal patterns; inter-quartile range; post-processing

## 1. Introduction

The leaf area index (LAI), defined as one-half of the total green leaf area per unit ground surface area [1], is the primary interface for the exchange of fluxes of energy, mass (e.g., water, nutrients, and $CO_2$) and momentum between the land surface and the planetary boundary layer. Since the LAI is an essential and a vital parameter in terrestrial ecosystems to characterize the structure and function of vegetation [1], it has been used in a variety of applications including plant photosynthesis modeling [2], calculation of potential evapotranspiration [3], biomass estimation [4], carbon source–sink studies [5], forest monitoring [6], vegetation phenology studies, extraction of plant biophysical parameters [7], and terrestrial ecosystem modeling at global and regional scales [8–10]. Spaceborne remote sensors with great spatiotemporal resolutions and larger area coverage capabilities provide an effective means to regularly monitor the changes at larger scales [11]. Several LAI products have been developed using the observations obtained from various earth observing satellites, including the Global Change Observation Mission–Climate (GCOM-C) [12], satellite products for change detection and carbon cycle assessment at the regional and global scales

(CYCLOPES) [13], NOAA-CDR-AVHRR-LAI [14], Copernicus Global Land Service LAI (CGLS-LAI) [15,16], ECOCLIMAP [17], MERRA-2 M2T1NXLND [18], GLOBCARBON [19], and MODIS [20–22]. The MODIS LAI products are widely used by researchers due to their higher spatiotemporal resolutions and greater temporal coverage [23]. The MODIS LAI algorithm retrieves the results using a lookup table inversion strategy based on the theories of the three-dimensional radiative transfer and the stochastic radiative transfer [20,24]. The operational algorithms include the main algorithm and the backup algorithms, which are based on the radiative transfer equation and the empirical relationship between canopy LAI and normalized difference vegetation index (NDVI) [21]. However, the remote sensing vegetation indices (VI) are often contaminated by long-term continuous clouds, shadows, snow, aerosols, and other artifacts significantly affecting the quality as well as the quantity of the data.

Several researchers have proposed various methods to improve and quality and quantity of the LAI products. These methods can be roughly categorized into three general groups: (1) the time-domain local filter methods, (2) the frequency-domain denoising methods, and (3) the function-based methods. Examples of time-domain local filter methods include the best index slope extraction (BISE) algorithm [25], the moving-average method, and the changing-weight filter [26]. However, the BISE algorithm requirements are too subjective and the effectiveness of the algorithm is often limited by individual skills and experienced strategies, e.g., how to determine the optimal sliding cycle VI acceptable percentage threshold and the regrowth percentage of the VI. Moreover, moving averages are likely to overestimate the non-growing season vegetation information if the curves are too smooth and, if sliding values are used as true values, the final fitted curves will deviate from the true information and result in the non-growing season information being overwritten. In addition, the changing-weight filter was intended to preserve the amplitude and shape of the VI timeseries, where a rule-based decision and a mathematical morphology algorithm are employed to identify the local maximum or minimum values and a three-point changing-weight convolution filter is employed to generate the new VI timeseries. However, the reconstruction results are not stable or reliable, with irregular fluctuations when successive atmospherically contaminated values occur. The approaches following the frequency-domain methods involve the Fourier-based fitting methods [27], wavelet transform methods [28], and harmonic analysis of timeseries (HANTS) methods [29]. The VI curves obtained by the use Fourier-based fitting methods are quite smooth. However, they are not suitable for irregular or asymmetric VI data since this method is critically dependent on symmetric sine and cosine functions. For non-smooth processes, the Fourier transform has limitations, one of which is that it is only possible to obtain which frequency components a signal contains in general but there being no knowledge of the moment when each component appears. Additionally, a small hormonic will lead to failing to capture quick changes in vegetation time-series. In contrast, a large harmonic will lead to emphasizing too much regional signal and over fluctuating. Furthermore, wavelet analysis replaces the infinite length trigonometric basis with a finite length decaying wavelet basis. Lu et al. [28] proposed a wavelet transform method to generate high-quality terrestrial MODIS products. However, it also reduced some reasonable high values, which limits its practical usage. Similarly, the HANTS method selects only the most significant frequencies in the time profiles and uses a least-square curve fitting procedure based on the harmonic components. However, it tends to overestimate the maximum NDVI values in the plateau of a timeseries and underestimates NDVI values when meeting several successive atmospherically contaminated values [30]. The other approaches, such as the asymmetric gaussian [31], general regression neural network [32], Whittaker smoother [33], weighted Whittaker smoother with dynamic parameter (wWHd) [34], Savitzky-Golay (S-G) filtering [35–37], and double logistic [38] methods follow the function-based methods. For S-G, it faces an ineluctable issue for its key parameter of half-width of the smoothing window and an integer specifying the degree of the smoothing polynomial [39]. The wWHd method, which is more stable and can capture a gradual change in vegetation even

for seriously contaminated vegetation timeseries. However, the wWHd is insensitive to the percentage of suitable points used to determine the critical weight and the minimum weight under different levels of random gaps, so it's uncertain when used in large-scale applications. In addition, the S-G, double-logistic (D-L), and asymmetric Gaussian function results may be misled by outliers, thus inevitably producing wrong peaks [34]. Moreover, these methods often ignore the identification of outliers when processing the MODIS data. By not removing outliers, any processing method will incorporate anomalous LAI values [34]. Although MODIS products provide a quality control (QC) documentation to explain the LAI value retrieved by retrieval algorithm, it is often too arbitrary to determine outliers based on QC documentation. The MODIS LAI algorithm primarily relies on a lookup-table-based procedure that utilizes the spectral information from the MODIS red (648 nm) and near-infrared (858 nm) bands. The success of the main algorithm is ensured within a specific range, i.e., where the distribution of values remains free from saturation. In such cases, the final value is determined by taking the mean value (QC < 32). Although this approach allows for the removal of points with poor performance and retrieval of LAI values without saturation and cloud interference, some reasonable values are also lost at the same time. Moreover, the poor performance of the MODIS LAI values retrieved using the main algorithm was also reported for certain types of landcover including evergreen needleleaf forests (ENFs), evergreen broadleaf forests (EBFs), and deciduous needleleaf forests (DNFs). Additionally, the quantity of the data was also affected by the presence of clouds, particularly over the EBFs located in tropical regions [40]. However, a strict regular pattern between the MODIS LAI and QC values cannot be observed. Additionally, the sudden spikes and valley peaks were also observed with both the back-up algorithm and the main algorithm in the presence of clouds or saturation of the MODIS LAI timeseries data, suggesting that the QC documentation was not sufficient to identify all the outliers in the dataset [36]. Therefore, it is important to explore new methods to identify and remove the outliers during the post-processing of MODIS data.

　　Typically, MODIS utilizes 4-day or 8-day maximum composite datasets and chooses the best available pixel values from multiple acquisitions of its Aqua and Terra sensors. This temporal composition approach allows reduction of the errors produced by the atmospheric factors and sensor variations. However, in the case of the LAI, MODIS retrieves daily LAI values for individual pixels without incorporating any temporal data; the lack of any temporal context in the retrieval process produces uncertainties in the results, eventually leading to increased noise levels in the timeseries [41,42]. Several studies [26,37] have reported the presence of outliers and a negative bias in the MODIS-derived LAI curves, and this may be attributed to atmospheric and technical challenges. Although utilization of the prior information to remove uncertainties and fill the data gaps is a common approach employed by the spatial data scientists [43], few similar studies have been conducted to improve the quality of the MODIS LAI. Several kinds of dynamic temporal data such as the landcover, phenological characteristics of vegetation, and interannual and seasonal changes may be utilized to improve the quality and the quantity of the MODIS LAI during the retrieval and post-processing stages. For instance, the LAI value in one pixel is similar for the same land type in the same DOY (day of the year) interannually. Therefore, the LAI deviation distance from the multi-year average can be important information for outlier judgment. In addition, the start of the season (SOS) and the end of the season (EOS) within a growing season (GS) are two commonly used metrics for assessing plant phenology and represent the patterns of vegetation photosynthetic activity and green leaf area annually [44–46]. These changes are reflected in the LAI curve as well, i.e., an ideal LAI curve decreases or increases monotonically with the changing seasons. Some specific landcover types such as EBFs located in the Amazon tropical forest have a specific LAI distribution pattern. The LAI over this region remains consistently high throughout the year [47]. The differences in the MODIS LAI over two different landcover types during the same period of time often produces outliers in the data [34,48]. In the light of these

challenges, the interannual and seasonal variation data for various land types can serve as effective information to identify and remove the outliers in the MODIS LAI dataset.

This study proposed a method to identify outliers in the MODIS LAI dataset during the post-processing stage through incorporating temporal dynamic information including the interannual and seasonal variations. Specifically, the objectives of this study are as follows: (1) To detect MODIS LAI outliers by including interannual and seasonal patterns in post-processing and (2) to test whether the method of identification of outliers can improve the quality of MODIS LAI. The results obtained from this study are also validated against 433 measurements collected from 52 global ground stations. Section 2 includes the description of methodology and datasets utilized in the study, and the results are described and discussed in Sections 3 and 4, respectively.

## 2. Materials and Methods

### 2.1. MODIS Datasets

MODIS LAI product: The LAI data in this study were sourced from the standard collection 6 MODIS LAI/FPAR product suite (MCD15A3Hv6) and was provided at a 500 m spatial resolution and 4-day temporal resolution with sinusoidal equal-area projection [49]. The global dataset (including the real-time updates) covered a temporal period from July 2002 to January 2022. Generally, the dataset included 92 composites per year, some of the data were missing due to technical issues. The missing data were produced using linear interpolation techniques.

MODIS phenology product: The phenology data were obtained the from the MODIS product MCD12Q2 (version 6.1) and had a spatial resolution of 500 m and a projection of sinusoidal equal-area. The reference data product had a temporal coverage from 2001 to 2021. It utilized the MODIS albedo product of MOD43B4 [50] to calculate the enhanced vegetation index (EVI) values to invert vegetation phenological timing. Specifically, the sliding window consisting of five consecutive time-phase EVI values was used to judge the continuous rising and falling intervals. When the extreme value and amplitude of the interval meet the given threshold conditions, the rising and falling interval is judged to be a growth cycle process and the one-year timeseries curve records, at most, two growth cycles. For each growth cycle, the piecewise logistic function was used to fit and the extreme point of curvature change was used to determine the beginning of growth, the midpoint of continuous EVI increase, the maturity, the peak, the midpoint of continuous EVI decrease, and the end of growth. Up to two growth cycles were recorded during a year [51]. The vegetation indices data could be retrieved from satellite observations and successfully applied to estimate canopy greenness, calculating vegetation phenology metrics at the landscape scale [52–54]. The detailed MCD12Q2 product information is introduced by the basic information about sample points in Supplementary Material Information S1.

MODIS Land Cover Map: The land cover data were obtained from the annual MODIS product MCD12Q1, which had a spatial resolution of 500 m and a temporal coverage from 2001 to 2021. In this product, the land cover types were obtained primarily from the supervised classification of the MODIS Terra and Aqua reflectance data, with additional subsequent processing of the supervised classification results, combined with a priori knowledge and auxiliary information, to further refine the category-specific land types. In this study, the MODIS International Geosphere-Biosphere Programme (IGBP) landcover classification scheme [55] was utilized. The spatial distribution of the landcover classification is displayed in Figure 1. The distribution of random samplings of MODIS LAI is introduced by the basic information about sample points in Supplementary Material Information S1.

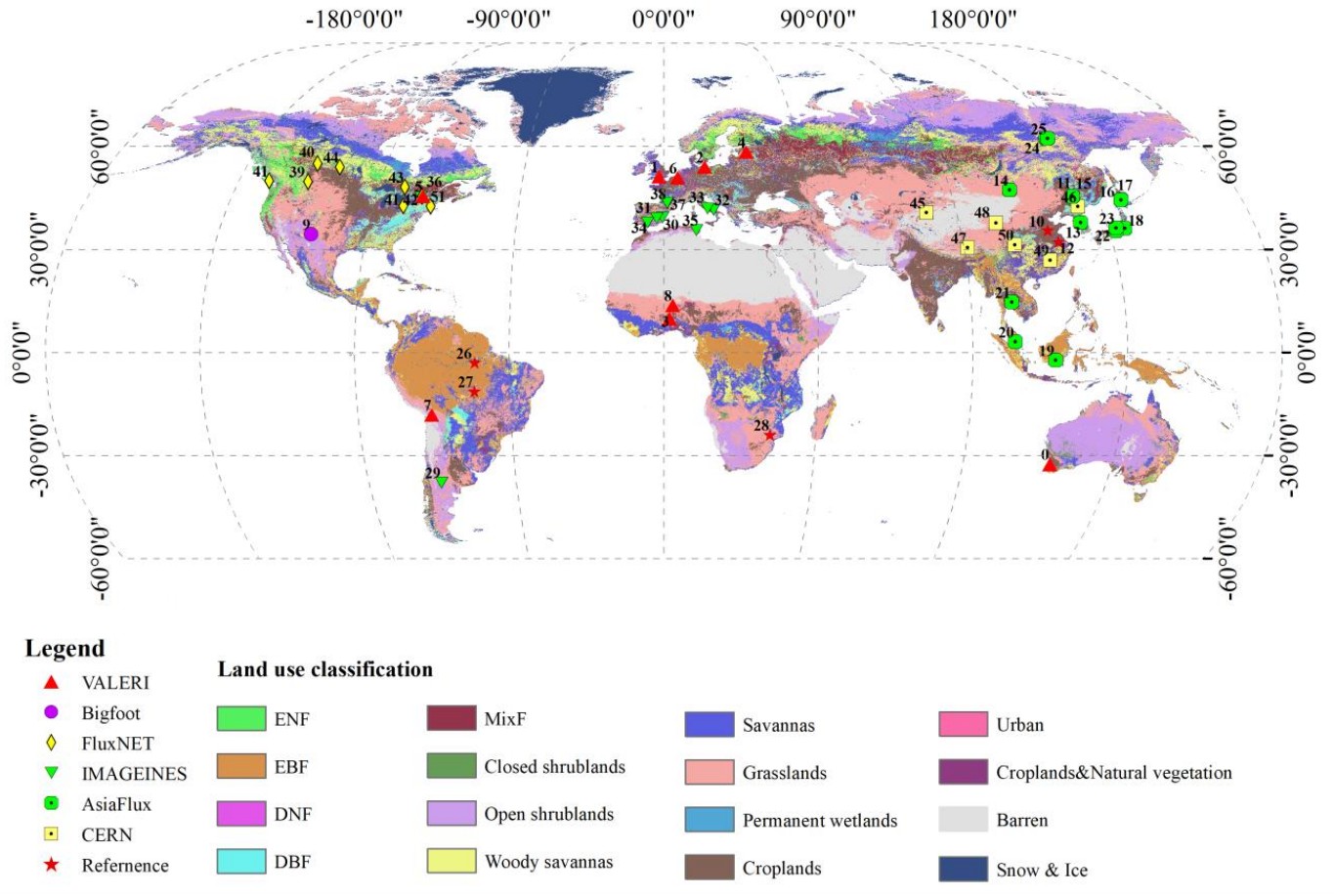

**Figure 1.** The distribution of the ground measurement LAI ENF, evergreen needleleaf forests; EBF, evergreen broadleaf forests; DNF, deciduous needleleaf forests; DBF, deciduous broadleaf forests; MixF, mixed forests.

*2.2. Ground Measurements LAI Data Collection*

The ground-based LAI measurements that were utilized to validate our results were obtained from several sources, including from the Validation of Land European Remote sensing Instruments (VALERI, http://w3.avignon.inra.fr/valeri/, accessed on 12 August 2023), the Implementing Multi-scale Agricultural Indicators Exploiting Sentinels, (IMAGINES, http://www.fp7-imagines.eu/, accessed on 12 August 2023), the Bigfoot (https://daac.ornl.gov/, accessed on 12 August 2023), the FLUXNET Canada (https://daac.ornl.gov/FLUXNET/g, accessed on 12 August 2023), the FLUXNET America (https://fluxnet.org/, accessed on 12 August 2023), the AsiaFlux (http://asiaflux.net/, accessed on 12 August 2023), and the Chinses Ecosystem Research Network (CERN, http://cernbio.ib.cas.cn/, accessed on 12 August 2023) projects. Moreover, the LAI measurements provided by some previous publications were also used in the study. More description about the datasets is provided in the following:

The VALERI and the Bigfoot LAI datasets: The VALERI program, started in the year 2000, is mainly supported by the Centre National d'Études Spatiales (CNES) and Institute National de la Recherche Agronomique (INRA), France. It performs global veracity checks on land remote sensing data products including LAI, FPAR, and FCover. The project incorporates a secondary sampling strategy based on elementary sampling units (ESU) for spatial scale conversion [42]. The Bigfoot project, supported by NASA's terrestrial ecology program, established the measurement sites in the United States (US) to validate the MODIS land products (e.g., land cover, LAI, FPAR, and NPP), and it remained operational from 1999 to 2003. It also employed a secondary sampling approach similar to that of the ESU in the VALERI program to obtain the measurements [56]. The VALERI project utilized a sample of

3 km × 3 km, whereas the Bigfoot project had a comparatively larger size, i.e., 5 km × 5 km. The scale of the ESU depended on the strategy adopted for scaling up and the resolution of the imagery, which typically ranged from 20 to 30 m. Moreover, the sampling method within the ESU varied based on the type of vegetation and the measuring instrument. For instance, for uniform and dense vegetation such as broadleaf and coniferous forests, the square mode was used and the LAI-2000 plant canopy analyzer instrument was utilized to obtain the measurements. The LAI-2000 plant canopy analyzer incorporated a "fisheye" optical sensor to measure the transmitted light from five angles above and below the canopy and used the canopy radiation transfer model (gap rate) to calculate the LAI. Additionally, the hemispherical canopy photography used a fisheye lens with a field angle closer or equal to 180°, and this method projected the entire hemisphere on the horizontal plane of the image. Conversely, for areas with a lower canopy height, including grassland, crops, and shrubs, the cross mode was proffered. Furthermore, the tracing radiation and architecture of canopies (TRAC) LAI detector instrument was utilized for sparse and discontinuous vegetation as it had a more intensive transect sampling mode with measurements at regular intervals along the transect. In addition, the LAI measurement map corresponds to the map derived from the determination of a transfer function between the reflectance values of the SPOT image acquired during (or around) the ground campaign and biophysical variable measurements; the derived biophysical variable maps consider zenith angle, cover fraction and tree height, etc. The sampling of each ESU (for information: a transect in the GPS file is composed of x ESUs) is based at least on twelve elementary images from above the understorey and from below the canopy. For each ESU, all the single point measurements were averaged to obtain a true representative value for the LAI. The number of the single point measurements (generally 12 for VALERI and 5 for Bigfoot) primarily depended on the scale of the ESU and the height of the vegetation canopy. In this study, we collected and utilized 46 effective measurements from 10 sites from both the VALERI and the Bigfoot projects.

IMAGINES: Since 2013, several field campaigns have been carried out to collect ground data for validating the satellite-derived biophysical products of the Copernicus global land service. The in situ measurements were acquired by the Earth Observation Laboratory (EOLAB) and local teams following the guidelines proposed by the Committee on Earth Observation Satellites Land Product Validation Group [57,58]. The ground-based measurements collected over the main vegetation types were upscaled by the EOLAB using high-resolution satellite imagery provided by the Satellite pour l'Observation de la Terre (SPOT) and LANDSAT 8 to generate high-resolution reference maps of several variables, including LAI, FPAR, and FCover. The EOLAB followed the protocols established by the VALERI project to generate the Ref. maps [59]. These reference maps are available to download from the IMAGINES website (http://www.fp7-imagines.eu/, accessed on 12 August 2023). In this paper, 70 measurements obtained from 10 sites were utilized to validate our results.

FLUXNET LAI data: The FLUXNET data that were used in this study were collected from several sources including FLUXNET-Canada, AsiaFlux, ChinaFlux (CERN), and FLUXNET-America (Harvard Forest station). FLUXNET-Canada, established by the FLUXNET-Canadian Research Network (FCRN) and the Canadian Carbon Program operated from 1993 to 2014. The data provided by the FCRN were obtained from measurements and simulations carried out by the site investigators. The in situ LAI data were measured using various instruments including the TRAC LAI detector, the LAI-2000 plant canopy analyzer, and the LI-3121 area meter. We collected 55 measurements from six FLUXNET-Canada sites to utilize in our study.

AsiaFlux was established with the aim to develop an easy-to-use open database providing the essential characteristics of the material exchanges (e.g., $CO_2$ and water) between individual sites (ecosystems) and the atmosphere. In this study, 100 measurements, mostly acquired using the hemispherical canopy photography approach, were obtained from 13 AsiaFlux sites and used for validation purposes.

CERN, with its 42 ecosystem research stations, covers the major vegetation types in China and has been providing observations for a long time. We collected 74 measurements from six CERN sites and used them in our study. The measurements were carried out following the grading and upscaling ground measurements method proposed in a previous study [57]. Moreover, the LAI surface measurements from the Harvard Forest Flux Observatory (42.538°N, 72.171°W), located in the Harvard Forest region of Massachusetts, USA were also used in the study.

LAI data collection from previously published research: We collected 55 measurements from previous publications and used them in the study. The details of these measurements are given in the following: Li et al. [60] evaluated the global product of the crop and grassland LAI in northern China against the LAI data measured from the fields of winter wheat in Liaocheng and Jining cities located in Shandong Province. The researchers collected the data through several campaigns at an image of size 1 km × 1 km. From these sample sites, they selected nine specific points and calculated the average value, which served as the validation reference data for their study.

Lu and Fan [61] conducted campaigns to obtain to field measurements fot the broadleaf and the mixed forests in an experimental forest site located at the Northeast Forestry University Maoershan Academic Center (NFUMA) using a TRAC meter. Moreover, Pan et al. [62] performed LAI measurements in 2017 and from early July to September 2018. They utilized the LAI-2200C canopy analyzer to sample 114 consecutive measurements within the sample plots, and the mean value of the sample plots was chosen as an accurate observation.

To deal with the insufficient LAI observation samples in tropical regions, the measurements were collected from three published studies that had their study areas located in these regions. Negrón et al. [47] measured the LAI based on hemispherical canopy photography that utilized a CI-110 digital plant canopy imager in Tapajos national forest (TNF). Pinto-Júnior et al. [63] used a photosynthetically active radiation sensor (model LI-190; LICOR Bioscience, Lincoln, NE, USA) coupled with a datalogger to obtain ground-based LAI measurements in a transitional forest located 50 km northeast of Sinop, Mato Grosso (SMG), Brazil. In addition, Dube et al. [64] measured LAI under clear sky conditions for both the dry and wet seasons using an LICOR-2200 plant canopy analyzer. The study was carried out in the Kruger National Park (KNP) in South Africa.

The ground-based LAI measurements collected from all the sources were screened and only the observations providing accurate geographical information and following the optical observation approach were selected. Finally, a total of 433 measurements from 52 sites fulfilled the criteria, and these measurements were used in the study. The geographic locations of the sites are displayed in Figure 1 and more details are provided in Table 1.

**Table 1.** Ground measurement LAI station information.

| No. | Name | Country | Lat (°) | Lon (°) | LC | Database |
|-----|------|---------|---------|---------|-----|----------|
| 0 | Camerons | Australia | −32.6160 | 116.2756 | EBF | |
| 1 | Chilbolton | England | 51.1642 | −1.4306 | Crops and forest | |
| 2 | Demmin | Germany | 53.8919 | 13.2072 | Crops and forest | |
| 3 | Donga | Benin | 9.7697 | 1.7453 | Grass | |
| 4 | Jarvselja | Estonia | 58.2994 | 27.2603 | Boreal forest | VALERI |
| 5 | Larose | Canada | 45.3806 | −75.2169 | Mixed forest | |
| 6 | Sonian | Belgium | 50.7683 | 4.4111 | Forest | |
| 7 | Turco | Bolivia | −18.2394 | −68.1933 | Shrub | |
| 8 | Wankama | Niger | 13.6450 | 2.6353 | Grass | |
| 9 | Sevi | USA | 34.3509 | −106.6899 | Grass | Bigfoot |
| 10 | Shandong | China | 35.4221 | 116.5292 | Crops | [60] |
| 11 | NFUMA | China | 45.3668 | 127.5915 | Forest | [61] |
| 12 | Nanjing | China | 32.0667 | 118.8512 | Forest | [62] |

**Table 1.** *Cont.*

| No. | Name | Country | Lat (°) | Lon (°) | LC | Database |
|---|---|---|---|---|---|---|
| 13 | GDK | Korea | 37.74888 | 127.1492 | Mixed forest | |
| 14 | KBU | Mongolia | 47.2140 | 108.7373 | Grassland | |
| 15 | LSH | China | 45.2786 | 127.5783 | Larch forest | |
| 16 | MBF | Japan | 44.3842 | 142.3186 | DBF | |
| 17 | MMF | Japan | 44.3219 | 142.2614 | Mixed forest | |
| 18 | MSE | Japan | 36.0540 | 140.0269 | paddy field | |
| 19 | PDF | Indonesia | −2.3450 | 114.0364 | Tropical forest | AsiaFlux |
| 20 | PSO | Indonesia | 2.9667 | 102.3000 | Tropical forest | |
| 21 | SKR | Thailand | 14.4924 | 101.9163 | Tropical EBF | |
| 22 | SMF | Japan | 35.2500 | 137.0667 | Mixed forest | |
| 23 | TKC | Japan | 36.1397 | 137.3708 | EBF | |
| 24 | YLF | Russia | 62.2550 | 129.2414 | Forest (larch) | |
| 25 | YPF | Russia | 62.2414 | 129.6506 | Pine forest | |
| 26 | TNF | Tapajos | −3.0170 | −54.9707 | Tropical forest | [47] |
| 27 | SMG | Brazil | −11.4125 | −55.3250 | Semi-DBF | [63] |
| 28 | KNP | Kruger | −24.0079 | 31.5489 | Semi-arid savanna | [64] |
| 29 | 25 May | Argentina | −37.9390 | −67.789 | Forest | |
| 30 | Albufera | Spain | 39.2744 | −0.3164 | ENF | |
| 31 | Barrax-LasTiesas | Spain | 39.0544 | −2.1007 | Crop | |
| 32 | Capitanata | Italy | 41.4637 | 15.4867 | Crop | |
| 33 | Collelongo | Italy | 41.8500 | 13.5900 | DBF | IMAGINES |
| 34 | LaReina-Cordoba-2 | Spain | 37.7929 | −4.8267 | Crops | |
| 35 | Merguellil | Tunisia | 35.5662 | 9.9122 | Crops | |
| 36 | Ottawa | Canada | 45.3056 | −75.7673 | Crops | |
| 37 | South West-2 | France | 43.4471 | 1.14510 | Crops | |
| 38 | South West-l | France | 43.5511 | 1.0889 | Crops | |
| 39 | AB-Lethbridge | Canada | 49.70919 | −112.9403 | Grassland | |
| 40 | AB-Western | Canada | 54.9538 | −112.467 | Mixed forest | |
| 41 | BC-Campbell River 2000 Douglas-fir | Canada | 49.8705 | −125.2909 | ENF | FLUXNET-Canada |
| 42 | ON-EPeatland-MerBleue | Canada | 45.4094 | −75.5187 | Shrubs | |
| 43 | Ontario Juvenile | Canada | 48.1330 | −81.6280 | Mixed forest | |
| 44 | SK-1975 Jack Pine | Canada | 53.8758 | −104.6453 | ENF | |
| 45 | Akesu | China | 40.6167 | 80.8500 | Cropland | |
| 46 | Changbaishan | China | 42.4000 | 128.1000 | Forest | |
| 47 | Dangxiong | China | 30.4690 | 91.0624 | Grassland | CERN |
| 48 | Haibei | China | 37.6167 | 101.3167 | Alpine meadow | |
| 49 | Qianyanzhou | China | 26.7475 | 115.0667 | EBF | |
| 50 | Yanting | China | 31.2667 | 105.4500 | Crop | |
| 51 | Harvard | USA | 42.5380 | −72.1710 | Forest | FLUXNET-America |

GDK, Gwangreung deciduous forest, Korea; KBU, Kherlenbayan Ulaan; LSH, Laoshan; MBF, Moshiri birch forest; MMF, Moshiri mixed forest; MSE, Mase paddy flux site; PDF, Palangkaraya drained forest; PSO, Pasoh Forest Reserve; SKR, Sakaerat; SMF, Seto mixed forest; TKC, Takayama evergreen coniferous forest; TNF, Tapajos National Forest; SMG, Sinop Mato Grosso; KNP, Kruger National Park; LC, Land cover.

### 2.3. LAI Outlier Identifying Methods

#### 2.3.1. Constructing LAI Residual Samples

We created random points on a global scale by using the sampling function given in the ArcGIS software (versions: 10.7.0.10450) and obtained 261 sample points for single-growth cycle vegetation and 17 sample points for double-growth cycle vegetation after excluding invalid points (those located over glaciers and rocks). The locations of the sample points are shown in Figure 1. Additionally, the type of sample is highlighted. The analyses of the

study were carried out over the locations of these points. The seasonal and interannual LAI residuals were calculated for the same types of landcover for each year from 2003 to 2021. Further, multi-year average and multi-year detrend analyses were carried out using the following set of equations:

$$M_{resi}{}^{yr,j}_{\text{mode},i} = \text{LAI}^{yr,j}_{\text{mode},i} - \overline{\sum_{2003}^{2021} \text{LAI}_{\text{mode},i}} \tag{1}$$

$$D_{resi}{}^{yr,j}_{\text{mode},i} = \text{LAI}^{yr,j}_{\text{mode},i} - (a_{\text{mode},i} \times yr, j + b_{\text{mode},i}) \tag{2}$$

where $M_{resi}{}^{yr,j}_{\text{mode},i}$ and $D_{resi}{}^{yr,j}_{\text{mode},i}$ represent the residuals of the land type $i$ and the year $j$ MODIS LAI distance form the multi-year average and multi-year detrend LAI, respectively, $\text{LAI}^{yr,j}_{\text{mode},i}$ is the MODIS LAI, and $\overline{\sum_{2003}^{2021} \text{LAI}_{\text{mode},i}}$ refers to the average LAI value for a specific land type from 2003 to 2021. Moreover, $a_{\text{mode},i}$ and $b_{\text{mode},i}$ represent the linear regression fitting slope and the intercept, respectively.

The seasonal LAI residuals were calculated and the seasons were defined based on the phenology of the vegetation during the start and end of the growing season, which corresponded to the Greenup and Dormancy time nodes provided in the MODIS MCD12Q2 product. The vegetation changes were divided into the growing season (GS) and the non-growing season (NGS). More details about the phenology nodes are provided in the Supplementary Materials. The regression values were obtained through fitting the LAI the GS using a quadratic polynomial equation. Finally, the seasonal LAI residual was calculated using Equation (3) given in the following:

$$S_{resi}{}^{yr,j} = \text{LAI}^{yr,j} - (a \times (yr, j)^2 + b \times (yr, j) + c) \tag{3}$$

where $S_{resi}{}^{yr,j}$ is the seasonal residual, $a$ is the coefficient of the quadratic term, $b$ is the coefficient of the first term, and $c$ is the constant term of quadratic polynomial fitting.

### 2.3.2. Using the Inter-Quartile Range to Identify LAI Outliers According to LAI Residuals

This study used the inter-quartile range (IQR) method to determine LAI outliers [65]. The method of identifying outliers using the IQR is different from other classical methods, such as the 3σ rule or the z-score method, which are primarily based on the assumption that the data have a normal distribution. However, the actual data often do not strictly obey the normal distribution. In addition, the 3σ rule or the z-score identify outliers based on the mean and the standard deviation of the data. However, the instability of the mean and standard deviation of the data may significantly affect the identification of the outliers. Therefore, the application of these two methods is not suitable for non-normally distributed data. On the other hand, the IQR is based on the actual data and identifies the outliers based on the quartile and the IQR, owing to the fact that the quartile has a certain degree of stability. As much as 25% of the data can be arbitrarily far away without greatly disturbing the quartile. Consequently, identifying outliers using the IQR is relatively objective [66]. The IQR-based method detects the outliers using the following set of equations:

$$outlier_{\max} = Q_{75} + x_1(Q_{75} - Q_{25}) \tag{4}$$

$$outlier_{\min} = Q_{25} - x_2(Q_{75} - Q_{25}) \tag{5}$$

where $Q_{25}Q_{25}$ and $Q_{75}$ are the first and third quantiles of the LAI residuals, respectively, and $x_1$ and $x_2$ are interval parameters.

Optimizing the IQR interval parameters in denoising by including interannual dynamics: We carried out sensitivity analysis to determine the optimal interval parameters $(x_1, x_2)$. The sensitivity analysis was based on the number and the accuracy rate of the denoising points. Therefore, it was important to determine whether the denoising points identified using Equations (4) and (5) were well aligned with the background value. Previous studies

have often calculated the multi-year mean of the LAI and treated that as a background value [67]. However, in this study, we utilized both the multi-year mean and the regression values under the same land class as a background value. If the LAI value fluctuates sharply within a complete GS and at the same time deviates from the background value, then the LAI is an abnormal value. Similarly, during the NGS, if the LAI significantly increases or decreases compared with the values of the preceding or succeeding periods and deviates from the background value, then the LAI is also an abnormal value. In addition, the normal Q–Q distribution diagram was plotted to validate the residuals obtained through the multi-year mean ($M_{\text{resi}_{\text{mode},i}}^{yr,j}$) and multi-year detrend methods ($D_{resi_{\text{mode},i}}^{yr,j}$), and the results displayed in Figure 2a,b indicate that although the residuals obtained using the two methods deviated from the normal distribution, they demonstrated noticeable symmetry. The results from methods such as the multi-year detrend, the multi-year mean, the intersection of multi-year detrend and multi-year mean (multi-year detrend+multi-year mean), and the union of multi-year detrend and multi-year mean (multi-year detrend | multi-year mean) showed that the number of denoising points gradually increased as the interval parameter decreased. The number of denoising points when reducing the intervals ($x_1,x_2$) from 1.5 to 0.5 using the four methods was observed as follows: multi-year detrend | multi-year mean (1289 to 4211) > multi-year mean (1192 to 3120) > multi-year detrend (783 to 3268) > multi-year detrend+multi-year mean (686 to 2177). Moreover, the number of denoising points fluctuated smoothly in the 1.0~1.5 intervals; however, a sharp increment was observed between the intervals 0.5 and 1.0 (Figure 2c). The correct rate increased from 0.5 to 1.1 and then decreased when the interval range was reduced from 1.1 to 1.5. At the interval 1.1, the total number of denoising points was 1186 and 966. The highest accuracies of 85.80% and 84.67% were achieved for the multi-year detrend and multi-year detrend | multi-year mean methods, respectively (Figure 2d).

Optimizing the IQR interval parameters in denoising by including seasonal dynamics: The normal Q–Q distribution diagram was constructed to verify the residual distribution within the GS ($S_{resi}^{yr,j}$). The results showed that the seasonal LAI residuals deviated from the normal distribution, particularly on the forward residuals side, and exhibited an apparent asymmetry (Figure 3a). Similar results were observed for the double-growth vegetation's residual distribution in each growth cycle (Figure 3d,g). The asymmetry of $S_{resi}^{yr,j}$ and the presence of outliers in the LAI during in the GS (usually low outliers) contributed to the asymmetry in the upper and the lower limit intervals ($x_1,x_2$) when detecting the outliers using Equations (4) and (5).

In this study, the dynamic interval parameters method was used to determine the appropriate upper and lower limit intervals. The method was based on two values, the count of the remaining effective LAI points after denoising in the GS and the regression correlation coefficient between these effective LAI points and their corresponding quadratic fitting values. The results show that the optimal combination was achieved when the upper limit interval was 1.5 and the lower limit interval was 0.3 for single-growth-cycle vegetation. In this case, the average count of the effective LAI points in the GS was 30.69 and the correlation coefficient was 0.75 (Figure 3b,c). In addition, the optimal combination was achieved when the upper and lower limit intervals were 1.5 and 0.2, respectively, for the first and second GS within dual-growth-cycle vegetation. The average number of effective LAI points in the first GS was 28.14, and the correlation coefficient was 0.73 (Figure 3e,f). Furthermore, for the second GS, the average count of effective LAI points was 27.57, and the correlation coefficient was 0.74 (Figure 3h,i).

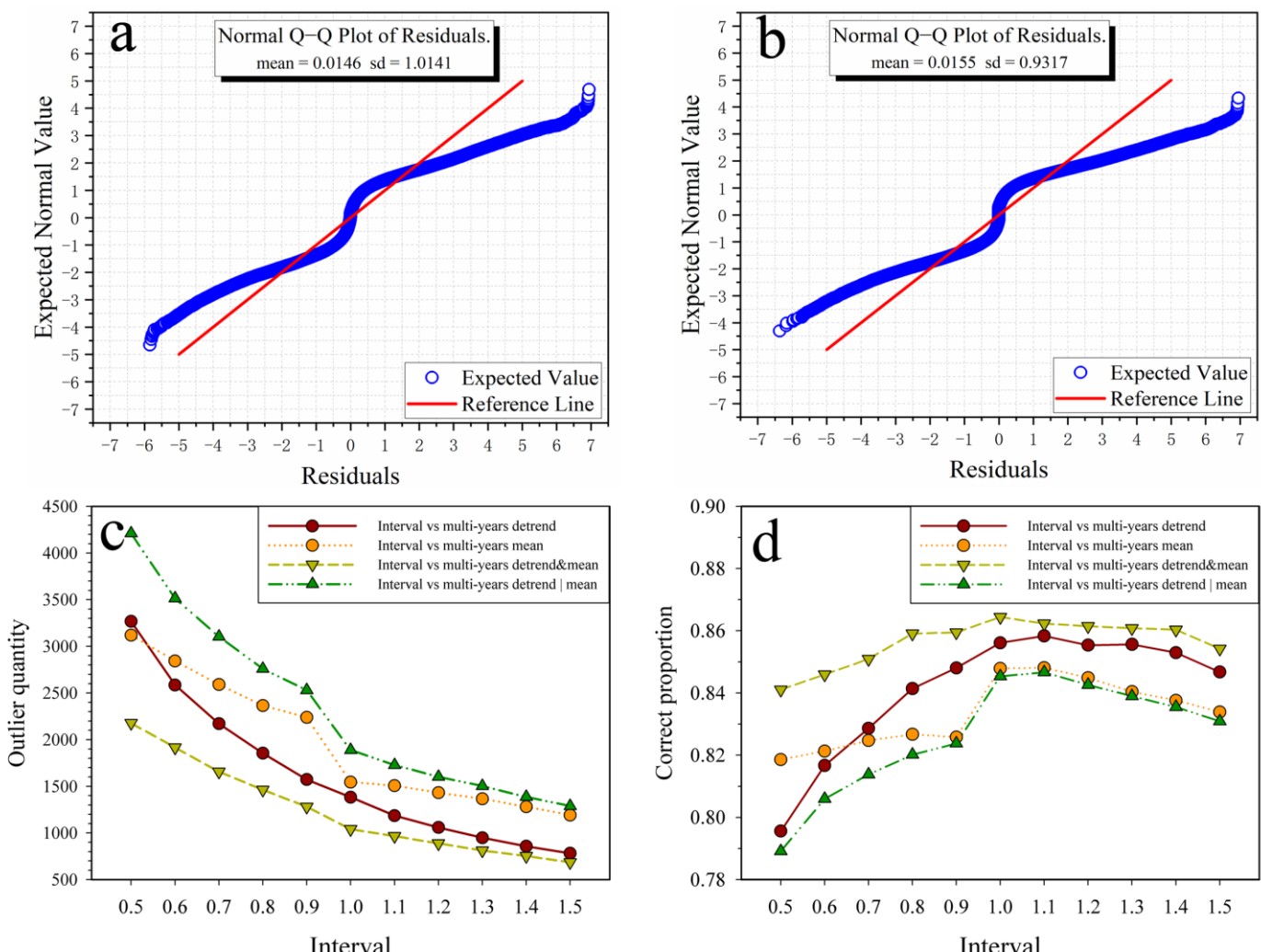

**Figure 2.** The normal Q–Q plot of residuals that were calculated through multi-year mean (**a**) and multi-year detrend denoising (**b**). (**c**) The number of denoising points identified as the intervals increase from 0.5 to 1.5. Correspondingly, (**d**) is the proportion of the correctly judged denoising points to the total number.

Additionally, the tropical forests that had a greater vegetation coverage experienced higher temperatures, levels of solar radiation, and precipitation and the LAI over these regions was very high throughout the year. However, in some areas there was no clear dividing point between GS and NGS, for example the samples No. 71, No. 86, etc. (Table S2). Therefore, we proposed a more robust way to identify the outliers based on the LAI threshold and distinguished seasonal patterns of the LAI for areas with and without apparent non-growing seasons. In detail, we calculate the third quantile of the whole year LAI ($Q_{75}$) and divide the year into four phases consisting of three months. If there exists an LAI tile that is greater than $Q_{75}$ in each stage, then there is no obvious non-growing season in this region and those LAI tiles that are less than the $Q_{75}$ in each phase were considered outliers. The detailed denoising results are introduced by the identification results of the MODIS LAI noise points in random sample points in Supplementary Material Information S2.

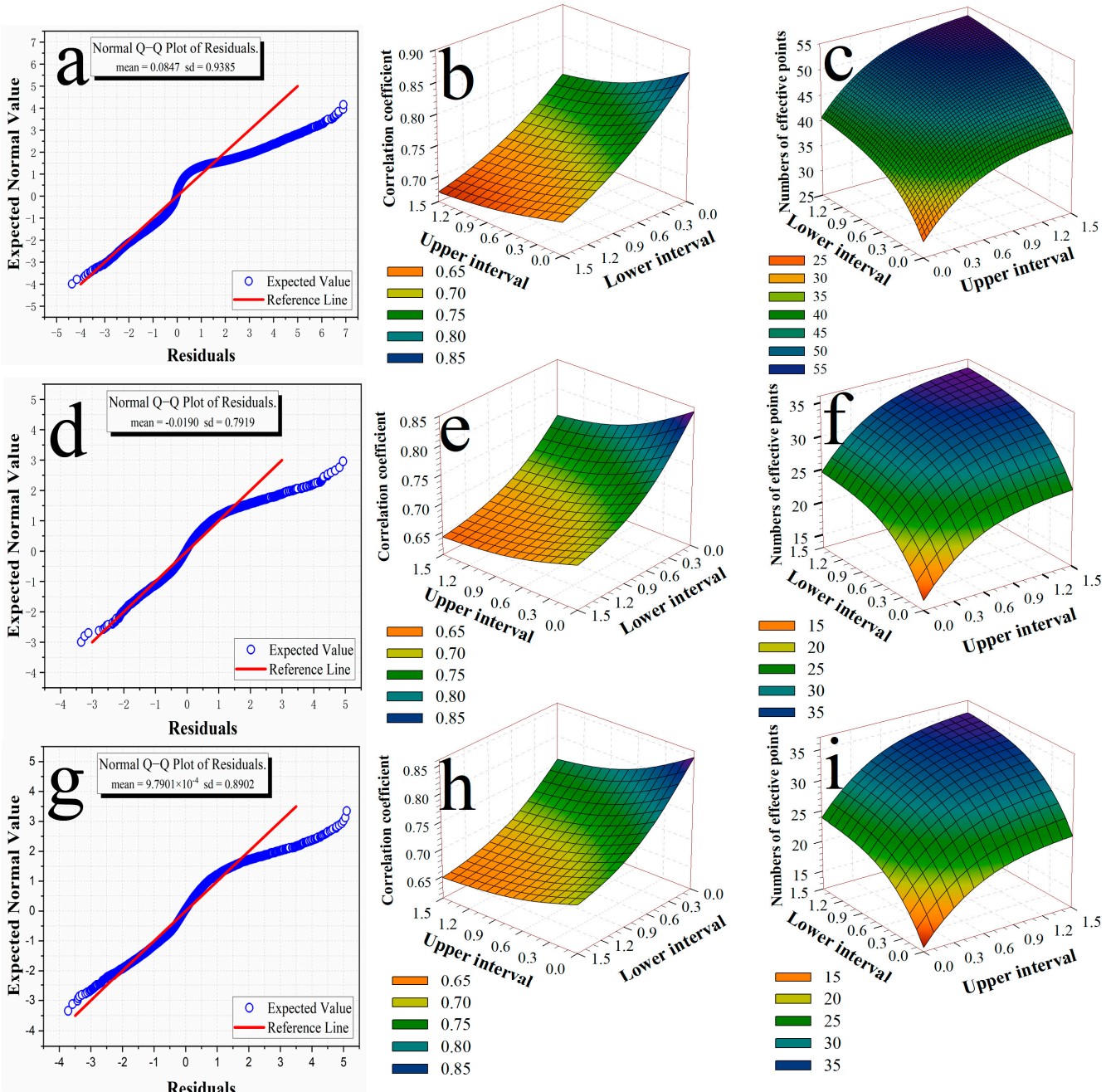

**Figure 3.** The normal Q–Q plot of $S_{resi}^{yr,j}$ and the criteria for the reasonable upper and lower interval parameters. Panels (**a,d,g**) represent the normal Q–Q plot of residuals for a single growth cycle and the first and the second GSs of dual-growth-cycle vegetation. Panels (**b,e,h**) are the correlation coefficients between the MODIS effective LAI points and contemporaneous polynomial fitting values in the GS of a single growth cycle and the first and the second GSs of dual-growth-cycle vegetation, respectively. Panels (**c,f,i**) suggest the numbers of MODIS effective points after eliminating the outliers in the GS of a single growth cycle and in the first and the second GSs of dual-growth-cycle vegetation, respectively.

*2.4. Validating MODIS LAI Outliers and Evaluating the Effect of the MODIS LAI in Post-Processing*

2.4.1. Outlier Identification in the MODIS LAI Dataset Using Ground-Based Measurements

Ground measurement (GM) LAI values were used to validate the outliers in the MODIS LAI dataset. When calculating the difference between the MODIS LAI and GM LAI records, the values that were significantly deviated from the GM LAI were treated

as outliers. These outliers included both unusually high and exceptionally low values compared with the GM LAI. We established the upper and the lower thresholds to identify the outliers using the following set of equations:

$$B_{i,\max} = delta_i + sdd \times M \tag{6}$$

$$B_{i,\max} = delta_i - sdd \times M \tag{7}$$

where $delta_i$ represents the difference between the MODIS LAI and GM LAI, $sdd$ is the standard deviation in the delta set, and $M$ stands for the multiple of $sdd$. When the $B_{i,\max}$ was greater than 0 or $B_{i,\min}$ was less than 0, the corresponding MODIS LAI is regarded as an outlier. In this study, we used 1.5σ and 1.0σ standard deviations to identify the outliers in the MODIS LAI dataset.

2.4.2. Evaluation of the Effect of MODIS LAI in Post-Processing

We identified the outliers in the post-processing of LAI data based on interannual and seasonal dynamics, and the results were validated against the GM LAI data. Several evaluation indices including the correlation coefficient ($R^2$), mean absolute error ($MAE$), root mean square error ($RMSE$), and standard deviation ($SD$) were utilized to assess the quality of the results. These indices were calculated using the following equations:

$$MAE(x,h) = \frac{1}{m} \sum_{i=1}^{m} \left| h(x^i) - y^i \right| \tag{8}$$

$$RMSE(x,h) = \sqrt{\frac{1}{m} \sum_{i=1}^{m} \left( h(x^i) - y^i \right)^2} \tag{9}$$

where $y^i$ and $h(x^i)$ represent the GM and the post-processing LAI measurements.

$$SD = \sqrt{\frac{1}{m} \sum_{i=1}^{m} \left( \Delta^i - \mu \right)^2} \tag{10}$$

where $\Delta^i$ indicates the difference between the post-processing LAI and GM LAI and $\mu$ is the average value of $\Delta^i$.

*2.5. Google Earth Engine Platform*

Processing global scale satellite datasets requires huge memory and efficient computing. The Google Earth Engine (GEE) is an emerging cloud-based platform that gives access to massive remote sensing datasets and efficiently processes them online. The GEE data catalog provides several geospatial datasets including the observations from spaceborne and airborne platforms, environmental variables, weather and climate forecasts, land cover, topography, and socioeconomic datasets [68]. The MODIS products utilized in this study were processed using the GEE.

**3. Results**

*3.1. Outliers of MODIS LAI Based on Ground Measurements*

Under the 1.5σ and 1.0σ standard deviations of the delta ($sdd$), a large difference was observed in the count of identified outliers. By applying the 1.5σ standard deviation, 35 abnormally high and 15 abnormally low outliers were identified in the MODIS LAI datasets and there was an $R^2$ of 0.82 between the satellite and ground-based measurements (Figure 4a). For the 1.0σ standard deviation, 54 high and 38 low outliers were detected in the MODIS dataset and the $R^2$ was improved to 0.89 (Figure 4b).

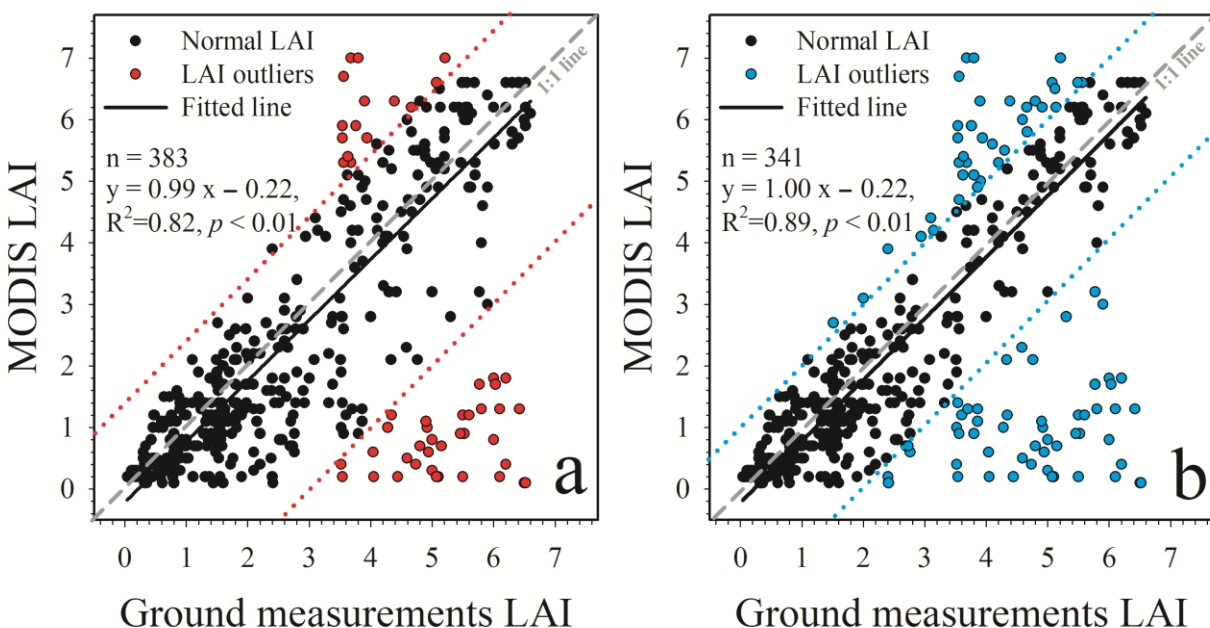

**Figure 4.** Outliers of MODIS LAI based on ground measurements. Graphs (**a,b**) show how the outliers were judged based on setting 1.5 and 1.0 times *sdd* respectively. The red and cyan dotted lines represent 1.5 and 1.0 times *sdd* judgment intervals, respectively.

*3.2. Identification of the Outliers by Including Interannual Patterns of LAI*

The timeseries comparison was carried out between the MODIS and the ground-based LAI measurements over several stations and the results are displayed in Figure 5. For the samples number 10, 11, 20, 41, 48, and 51, the multi-year mean denoising (MYMD) method detected 20, 23, 36, 7, 29, and 25 noise points, respectively. Correspondingly, 23, 24, 35, 8, 37, and 24 noise points were identified by the multi-year detrend denoising (MYDD) method (Figure 5a–g). In total, 1126 and 1076 noise points were identified for all the stations under based on the MYMD and the MYDD methods, respectively. Moreover, the growing season denoising (GSD) detected 38, 28, 54, 50, 51,37, and 46 noise points, over the stations with numbers 10, 11, 20, 40, 41, 48, and 51, respectively (Figure 5a–g). Overall, the GSD-based method identified a total of 2084 noise points for all the stations. The results showed that the $Q_{75}$ threshold denoising ($Q_{75}TD$) only worked for specific stations, i.e., 264 noise points were identified for station number 20 (Figure 5c). More details including the count of noise points over each station is provided in the Supplementary Material.

Figure 6 shows the validation of the MOIDS-derived LAI measurements against 433 samples obtained from 52 ground stations. The MYMD method identified 25 outliers, of which, 3 that were located in the LSH and KBU sites were significantly overestimated and 4 over the stations located in Shandong, Yanting, and KBU were notably underestimated compared with the ground-based measurements. The two datasets exhibited an $R^2$ of 0.51 (Figure 6a). After removing the outliers, the correlation between the MODIS and ground-based LAI measurements was improved ($R^2 = 0.53$) (Figure 6b). The MYDD method identified 18 outliers, among which 4 outliers were overestimated and 3 outliers were underestimated compared with the ground-based measurements. The overestimated outliers were observed over the LSH and the KBU stations. After removing outliers, the MYDD method also showed an $R^2$ of 0.53, which was similar to that for the MYMD method (Figure 6c). In addition, a total of 29 outliers were identified in the MODIS LAI dataset through employing the MYMD+MYDD method, out of which, 5 and 4 outliers were significantly over and underestimated, respectively. The correlation coefficient between the MODIS LAI after applying the MYDD+MYMD method and GM LAI was measured to be 0.54 (Figure 6d). Further details, including the identified outliers over each of the sites, are listed in the Supplementary Materials.

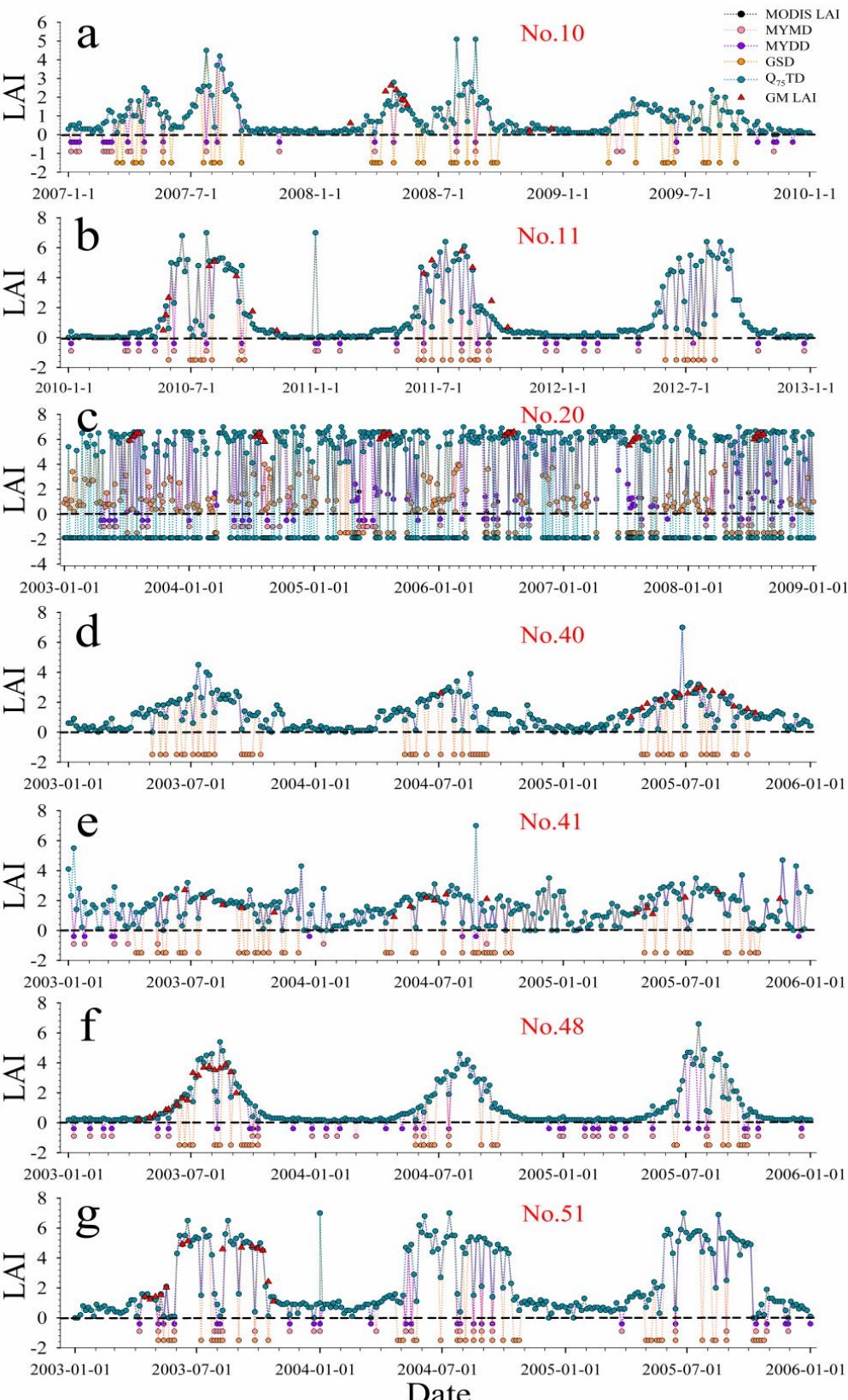

**Figure 5.** Timeseries plots of the MODIS LAI (black), multi-year detrend denoise (purple), multi-year mean denoise (pink), growing season denoise (orange), $Q_{75}$ threshold denoising (dark cyan), and ground measurement LAI (red). (**a**–**g**) represent number 10, 11, 20, 40, 41, 48 and 51 ground measurement LAI station.The outliers identified via multi-year detrend, multi-year mean, growing season, and $Q_{75}$ threshold denoising were marked with −0.5, −1.0, −1.5, and −2.0.

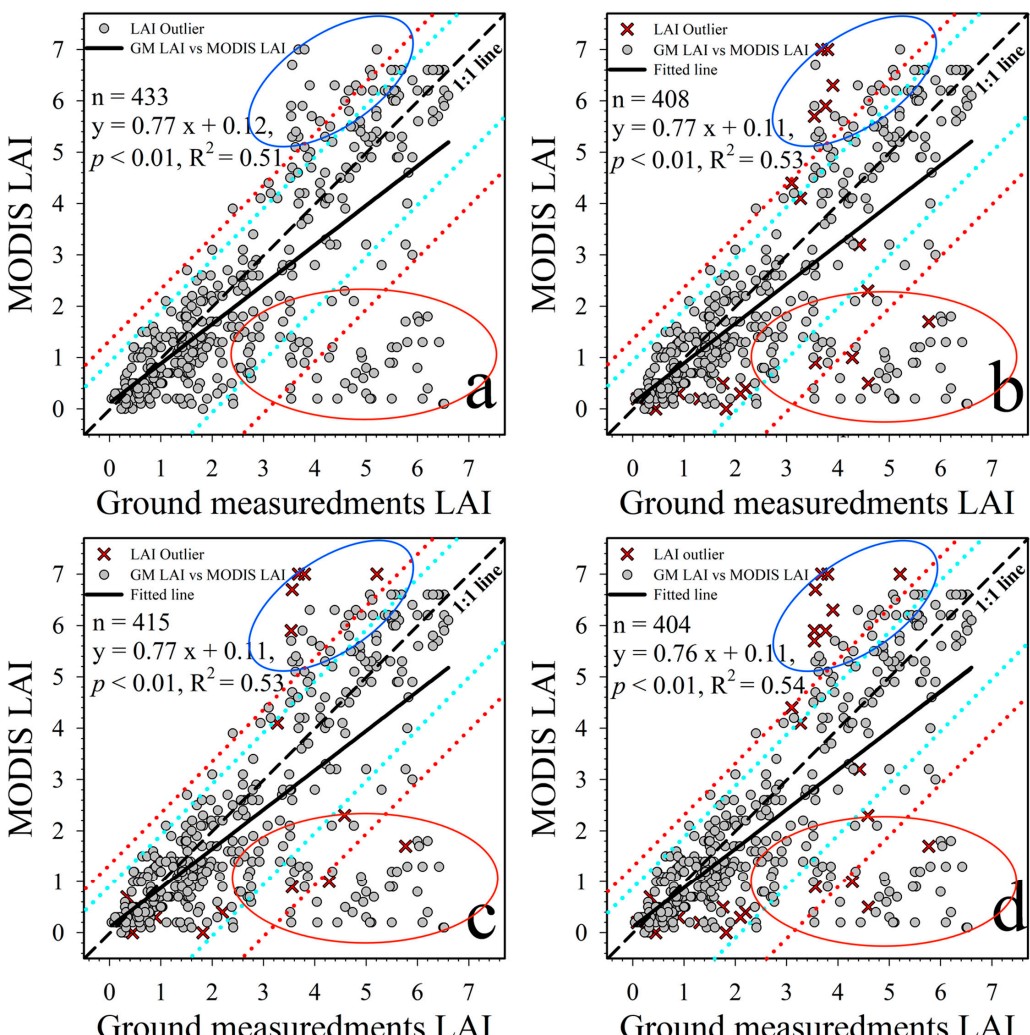

**Figure 6.** Identification of LAI outliers by including interannual dynamics and the comparison with the ground measurement LAI. Panel (**a**) represents the comparison of MODIS LAI and ground measurement LAI; (**b–d**) shows the comparison of MODIS LAI after MYMD, MYDD, and MYMD+MYDD, respectively, with ground measurement LAI. (The red cross represents the identified LAI outliers. The red and cyan dotted lines represent 1.5 and 1.0 times *sdd* judgment intervals, respectively. Points in the blue circle indicate that MODIS LAI is significantly larger than the ground measurements LAI, and points in the red circle show that MODIS LAI is significantly smaller than the ground measurements LAI.

Table 2 presents a comparison result between the number of MODIS LAI outliers identified through different denoising methods when applying 1.5σ and 1.0σ *sdd*. Under the 1.5σ standard deviation, the MYMD, MYDD, and MYMD+MYDD identified 8, 7, and 11 outliers, respectively. This corresponds to the proportions of identified outliers at 16%, 14%, and 22%, respectively. Among these, five, five, and eight outliers were instances where the MODIS LAI significantly exceeded the GM LAI values. Such cases were found at the SMG, Nanjing, LSH, Albufera, and MBF sites. Conversely, three, two, and three outliers were detected where the MODIS LAI values were notably lower than the GM LAI observed at the NFUMA and Harvard sites. In the case of 1.0σ *sdd*, the MYMD, MYDD, and MYMD+MYDD detected 11, 9, and 14 outliers, with proportions of 11.96%, 9.78%, and 15.22%, respectively. The results included six, five, and nine outliers where the MODIS LAI was significantly overestimated and five, four, and five outliers where the MODIS LAI was underestimated compared with the GM LAI for the MYMD, MYDD, and MYMD+MYDD methods, respectively. The overestimated results were observed at the Nanjing, LSH, MBF,

SMG, SK-1975 Jack Pine, and Albufera sites and the underestimated results were found at the NFUMA, LSH, Collelongo, and Harvard sites. More details about the validation results are given in the Supplementary Materials.

**Table 2.** Comparing the number of MODIS LAI outliers identified by different denoising types with the MODIS LAI outliers judged by 1.5 and 1.0 times *sdd*.

| Denoising Steps | 1.5 Times *sdd* | | 1.0 Times *sdd* | |
| --- | --- | --- | --- | --- |
| | Total Outliers | Identified Outliers | Total Outliers | Identified Outliers |
| Step1: MYMD | 50 | 8 (16.00%) | 92 | 11 (11.96%) |
| Step2: MYDD | 50 | 7 (14.00%) | 92 | 9 (9.78%) |
| Step3: MYMD+MYDD | 50 | 11 (22.00%) | 92 | 14 (15.22%) |
| Step4: GSD | 50 | 33 (66.00%) | 92 | 51 (55.43%) |
| Step5: $Q_{75}TD$ | 50 | 19 (38.00%) | 92 | 20 (21.74%) |
| Step6: GSD+$Q_{75}TD$ | 50 | 38 (76.00%) | 92 | 56 (60.87%) |
| Final (steps: 1–6) | 50 | 46 (92.00%) | 92 | 65 (70.65%) |

### 3.3. Identification of Outliers by Including Seasonal Patterns of LAI

3.3.1. Identification of Outliers in Areas with Growing and Non-Growing Seasons

The outliers in the MODIS LAI were identified through incorporation of seasonal dynamics and the results are displayed in Figure 7. The seasonal denoising method identified 110 outliers, among them, 4 outliers were instances where the MODIS LAI values were significantly overestimated and 49 outliers were underestimated compared with the GM LAI measurements. The overestimated values were observed at the Donga and Dangxiong sites. Moreover, the satellite-derived and ground-based LAI measurements exhibited a good correlation, i.e., $R^2 = 0.78$ (Figure 7).

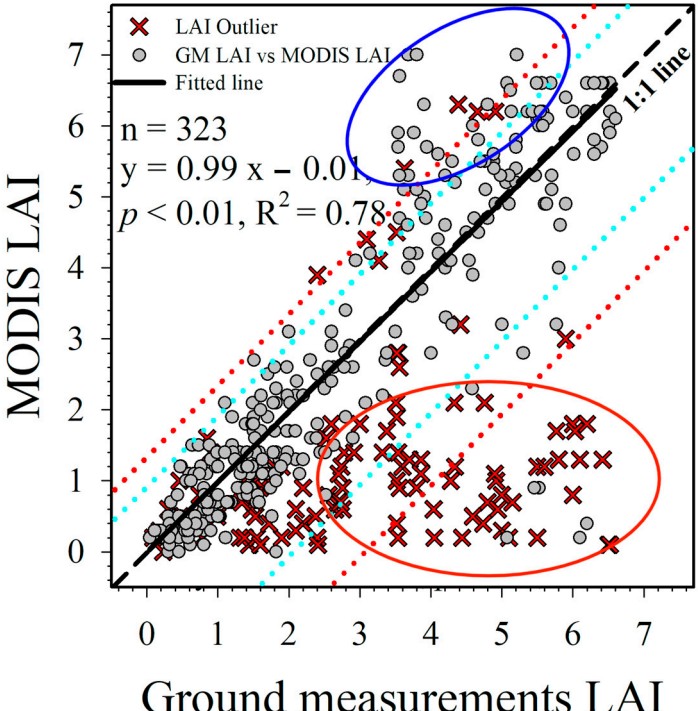

**Figure 7.** Identification of LAI outliers when including growing seasonal information, and the comparison of MODIS LAI after growing season denoising and ground measurement LAI. The red and cyan dotted lines represent 1.5 and 1.0 times *sdd* judgment intervals, respectively.

Furthermore, after applying 1.5σ and 1.0σ times *sdds* between the MODIS LAI and GM LAI as the criteria, the GSD identified 33 and 55 outliers, respectively. The results

obtained through this method showed that the MODIS LAI measurements were generally underestimated compared with the GM LAI observations.

### 3.3.2. Identification of Outliers in Areas without Non-Growing Seasons

The $Q_{75}$ threshold denoising method identified 23 outliers, and a general underestimation was observed in the MODIS LAI values compared with the ground-based observations. The two datasets exhibited an $R^2$ of 0.67 (Figure 8a). Moreover, the combined utilization of both the $Q_{75}$ and season-based methods detected 122 outliers; upon removing these outliers, a strong correlation (an $R^2$ value of 0.88) was achieved between the satellite and ground station measurements. (Figure 8b).

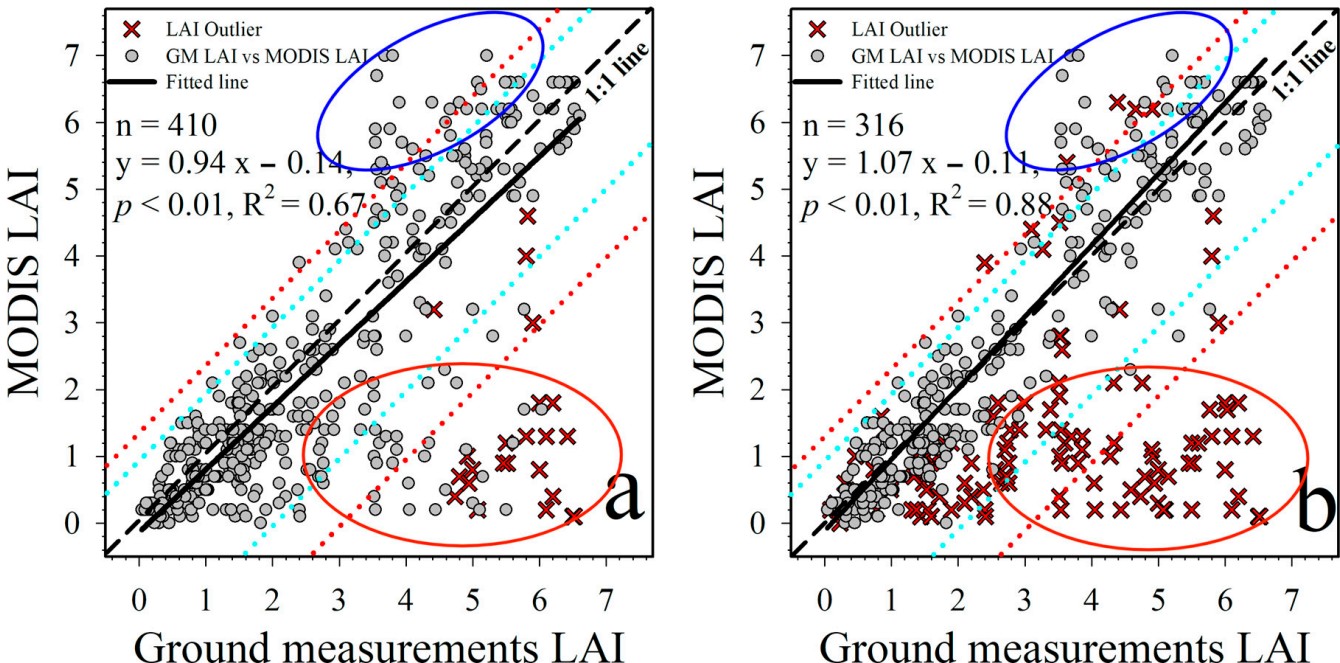

**Figure 8.** Identification of LAI outliers by including seasonal dynamics, and the comparison of MODIS LAI after season denoising and ground measurement LAI. The red and cyan dotted lines represent 1.5 and 1.0 times *sdd* judgment intervals, respectively. (**a**) and (**b**) are the comparison of MODIS LAI after $Q_{75}$TD and GSD + $Q_{75}$TD, respectively, with ground measurements LAI.

In the case where outliers were defined by 1.5σ sdd, the $Q_{75}$ TD and the GSD+$Q_{75}$ TD methods could identify 19 and 38 outliers, respectively. The proportion of identified outliers was 38.00% and 76.00%. Moreover, at 1.0σ sdd, the $Q_{75}$ TD and GSD+$Q_{75}$ TD methods detected 20 and 56 outliers, with the proportions 21.74% and 60.87%, respectively. The results obtained through these methods showed that the MODIS LAI values were mostly overestimated compared with the ground-based measurements. Further details including the validation results are given in the Supplementary Materials.

### 3.4. Identification of Outliers by Including Both Interannual and Seasonal Patterns of LAI

Considering both interannual and seasonal patterns for the LAI, a total of 133 outliers were identified in the MODIS LAI dataset. After removing these outliers, a strong correlation (with an $R^2$ values 0.90) was exhibited between the MODIS and ground-based LAI measurements (Figure 9).

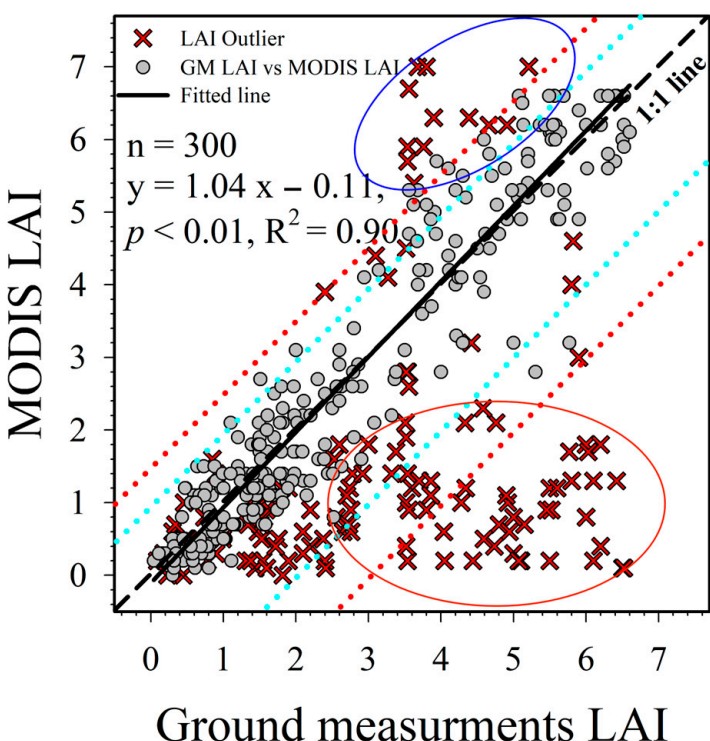

**Figure 9.** Identification of LAI outliers by including both interannual and seasonal dynamics, and the comparison with the ground measurement LAI. The red and cyan dotted lines represent 1.5 and 1.0 times *sdd* judgment intervals, respectively.

This method detected 46 and 65 outliers in the MODIS LAI dataset by applying standard deviations of $1.5\sigma$ and $1.0\sigma$, respectively. The results obtained from this method showed that, relative to the ground station LAI measurements, the MODIS LAI values were mostly underestimated; however, in some instances, overestimation in the MODIS LAI measurements was also observed. More details about the validation results are given in the Supplementary Materials.

The other evaluation indicators including the MAE, RMSE, and SD were also calculated for each denoising method, and the results are given in Table 3. The MAE, RMSE, and SD between the MODIS and GM LAI measurements were 1.01, 1.59, and 1.51, respectively. The MAE, RMSE, and SD values decreased after eliminating LAI outliers detected based on the interannual dynamics. Remarkably, the MAE, RMSE, and SD values between MODIS LAI after eliminating outliers identified by including interannual and seasonal dynamics and true LAI were 0.50, 0.68, and 0.68, respectively (Table 3).

**Table 3.** Evaluation results of MODIS LAI in post-processing.

| Denoising Steps | Ground Measurement LAI | | |
| --- | --- | --- | --- |
| | MAE | RMSE | SD |
| Original MODIS LAI | 1.01 | 1.59 | 1.51 |
| Step1: MYMD | 0.97 | 1.56 | 1.48 |
| Step2: MYDD | 0.98 | 1.56 | 1.47 |
| Step3: MYMD + MYDD | 0.97 | 1.56 | 1.47 |
| Step4: GSD | 0.63 | 1.02 | 1.02 |
| Step4: $Q_{75}$ TD | 0.83 | 1.23 | 1.22 |
| Step5: GSD+$Q_{75}$ TD | 0.55 | 0.78 | 0.78 |
| Final (Steps: 1–6) | 0.50 | 0.68 | 0.68 |

## 4. Discussion

### 4.1. Post-Processing Is Important to Improving the MODIS LAI Database

The processing of MODIS LAI outliers is the first and the most critical step. The MODIS LAI outliers are mainly produced by the following factors: firstly, the uncertainty of land surface classification; secondly, the ambiguity of land surface reflectance data; and thirdly, the uncertainty of the MODIS algorithm [22] (e.g., with saturation, the main algorithm may quickly bring higher LAI values in the MODIS observations and the backup algorithm can lead to both lower and higher values in most situations [36]). MODIS outliers can be identified by incorporating temporal dynamics into the post-processing stage. Considering the MODIS LAI interannual change is helpful for MODIS LAI outlier recognition [37]. Significantly higher (such as those on the dates 21 January 2005, 24 May 2005, and 13 March 2006 for sample No. 162) and lower (such as those on the dates 5 January 2003, 3 March 2003, etc., of sample No. 68) noisy points that deviated from the reasonable values were detected using the inter-quartile range (IQR). In fact, the evaluation results at the LSH, KBU, Shandong, and Yanting stations confirmed the identification of the over and undervalued LAI outliers based on interannual dynamics. Previous studies have proven that post-processing methods improved LAI products and made them more consistent and continuous [34,36,37,69], which further suggested that the shape of the sudden peaks and valleys in adjacent time periods belongs to the outlier category. In addition, several studies have proven that climate change can produce changes in phenology [70,71]. When constructing residuals based on the multi-year LAI values, it is important to take into account the interannual variation trend of the LAI for the same day of the year (DOY) for every year. This approach is useful in detecting outliers, as evident by the results of this study (i.e., the date 17 February 2003 in the sample dataset, denoted as No. 0).

Seasonal patterns are important factors to identify LAI outliers in post-processing. The results from this study showed that the proportion of outliers identified by the GSD was greater than that identified by other methods. Phenology, such as the timing of the SOS and EOS, the maximum LAI, and the amplitude of LAI (i.e., the difference between the maximum and the minimum LAIs in a growth cycle), is an essential metric for distinguishing the change in land vegetation cover [72]. There will be an apparent change in deciduous vegetation's regulation loop of "greenup–peak–greendown–senescence–dormancy–greenup" within a full growth cycle. Occasionally, when there is a vegetated surface with a dark (wet) soil background, the vegetation index also can be positively biased [43,50]. However, the contaminated remote sensing VI is mostly negatively biased, especially in the GS [37,73]. This characteristic leads to asymmetric judgment intervals using the IQR method to perform LAI anomaly judgments in GS. The advantage of using quadratic polynomial fitting to construct residuals to identify outliers in the GS is that the quadratic polynomial has only one peak, so it can well reflect the variation trend of LAI in a single growth cycle. From the perspective of different land use types, some terrestrial types have a clear phase transition from the NGS to the GS, such as grassland, MixF, ENF, savanna, DBF, open shrub, DNF, etc. [26], in which the GSD method can remove the outliers well, such as the results from samples No. 0, No. 74, No. 162, and No. 260. In addition, the evaluation results at each ground measured station, such as Demmin, Sonian, and GDK, confirmed the applicability of identifying abnormally low values in the GS based on seasonal dynamics. Additionally, the identification of the outliers improved within each GS over two growth cycles, as observed at the Shandong station. However, the seasonal patterns of the woody savanna type were not obvious in the tropical regions, as evidenced by samples No. 146, No. 181, and No. 221. In contrast, distinct seasonal characteristics were observed in the woody savanna type landcover located in the non-tropical regions, such as No. 18, No. 35, and No. 53 (see Information S2 in Supplementary Material). In addition, sudden fluctuations (drastic spikes or down) of the LAI in the GS are also abnormal for evergreen plants. Nonetheless, the LAI decreases were significantly affected by the dry season for monsoon forests; however, this process is often relatively traceable.

There were four outliers that remained unidentified even by including both interannual and seasonal patterns when considering 1.5σ *sdd*. These outliers were located at the MMF, LSH, Nanjing, and TNF sites. For those sites, although the upscaling strategy was utilized to possess the observations, the presence of uncertainty arose from a spatial scale mismatch between the satellite and ground-based observations. Moreover, the MMF site was located in a mixed forest and MODIS LAI outliers were assessed on 5 August 2005, during a growth period. However, comparing the MODIS LAI tile with that of the moments before and after being in the same level, it is not actually an outlier, even though it was judged as an outlier by 1.5 times *sdd*. The main reason why it is judged as an outlier is the difference between this site's observation values and the MODIS LAI observation values. Similarly, the MODIS LAI at the LSH (on 24 July 2008) and Nanjing (on 20 July 2017) sites, located in deciduous forests, were assessed during the growing seasons and no outliers were detected. In addition, the TNF site was in a tropical forest where the perennial LAI was at a high level and the MODIS LAI datapoint on 13 January 2004 was not an outlier, which agreed with the results obtained by including interannual and seasonal patterns. Furthermore, when the MODIS LAI outliers were identified by 1.0 sigma standard deviation, it was found that 27 records were not detected by the temporal pattern method proposed in this study. When the annual LAI level was high at a site, the absolute error between the MODIS and the ground-based LAI observations increased. In this case, the outliers identified under a 1.0 sigma standard deviation may overestimate the count of outliers.

In addition, the average of the original MODIS LAI across 433 recording points was 2.20; however, it increased to 2.44 after the post-processing, showing that the original MODIS LAI values were underestimated by 10%. Furthermore, the correlation coefficient between the MODIS after removing the outliers and GM LAI was 0.90, which was better than previous studies, i.e., 0.78 [36] and 0.80 [32], although these adopted S-G and general neural networks in post-processing, they ignored outliers' interference and thus inevitably produced wrong peaks, which is emphasized in the research of Kong et al. [34]. However, although there are 87 and 68 outliers respectively identified by total denoising, which is not in accordance with the outliers judged by 1.5 and 1.0 times *sdd*, most of the identified outliers from the MODIS LAI are significantly smaller than the GM LAI datapoints in some sites. The reason for this was that the standard deviation of *sdd* was 1.51 m$^2\cdot$m$^{-2}$; however, the variation interval of the LAI in the low vegetation cover area could not reach this standard deviation range, so the LAI outliers in this area could not be determined by a standard deviation method. In reality, a small number of noise points are misjudged as outliers in either the non-growth period or the growth period. Nevertheless, MODIS LAI has a large sample size (92) throughout the year, removing these outliers can be achieved by other reasonable means, such as double-logistic [38] and S-G filtering [35], etc. Therefore, the small number of misjudged outliers did not affect the overall effect.

### 4.2. Outlier Identification in Areas without Apparent Non-Growing Seasons

The quality of the phenology products is derived from the timeseries of the MODIS observed land surface greenness, which is influenced by the timeseries of the 2-band enhanced vegetation index calculated from the MODIS nadir bidirectional reflectance distribution function adjusted surface reflectance [51], which affects the GSD. From the respective of different land cover types, the MYMD, MYDD, and GSD can identify outliers in deciduous forest, such as the sample points in No. 14, No. 74, No. 104, and No. 295; in evergreen forest, such as the sample points in No. 20, No. 101, and No. 200; and in other types such as grassland, mixed forest, woody savanna, savanna, shrub, wetland, and crops, correspond to the sample points in No. 0, No. 6, No. 9, No. 28, No. 84, No. 125, and No. 214. However, those methods are not effective for particular types. For example, the bad characteristic of the phenology phase is still being determined because of persistent cloud cover and suboptimal atmospheric conditions in tropical regions. As a result, the effect of the GSD in this region is less effective. As shown in the No. 26 of Figure S3 of the Supplementary Materials, the outlier identification was poor from 1 September 2003

to 1 March 2004 and 1 November 2004 to 1 April 2005 using the GSD methods. This was likely to be caused by the presence of no significant phase shift from the non-growth to the growth season for some land types, i.e., EBF and woody savanna in tropical zones. Even though the denoising proposed in this study was capable of identifying outliers, all the outliers could not be detected over the EBF land type.

The IQR-based outlier identification method is based on the assumption that the number of contaminated LAI points is less than that of the normal LAI points. However, several MODIS pixels retrieved by the backup algorithm throughout the year are contaminated by clouds specifically over the EBFs located in tropical regions. Yuan et al. reported that the MODIS LAI values were significantly underestimated relative to the actual LAI measurements in the tropical EBF land type in different seasons [36]. Before setting a threshold to identify LAI outliers, the LAI characteristics for each season must be determined throughout the year. For the points that generally exhibit a higher level of LAI during each season throughout the year, such as the samples No. 2, No. 33, and No. 71 (Figure S2 in the Supplementary Material shows the results of other samples denoising results), the outliers in these samples can be effectively removed by setting appropriate thresholds. However, the LAI may also have some obvious seasonal variations for some EBFs [26], such as the sample No. 128, where the LAI values from November to March the next year were significantly lower than those from March to November, and the $Q_{75}$TD method was not applicable in this case. In addition, previous studies also reported that the LAI values in the tropical rain forest during a growing season were very high [32,47,67]. Juarez et al. [47] proposed a method to improve the estimate of the leaf area index based on the histogram analysis of hemispherical photographs, and LAI changes from June 2000 to May 2003 ranged from 3.72 to 6.48, with an average value of 4.92 through the continuous monitoring of the Amazon tropical forest, which supported the evidence that an excessively low LAI was an outlier in an area without a non-growing season for the EBF type land cover. In our study, the evaluation results of LAI outliers in the PDF, PSO, SKR, TNF, and SMG stations also confirmed the validity of identifying low-LAI outliers based on the $Q_{75}$ TD method.

The LAI characteristics in the woody savanna land cover type could be summarized into three types: (1) It has prominent seasonal variation characteristics, such as samples No. 9, No. 18, No. 38, and No. 53, located in subtropical, temperate, or high-latitude arctic regions. Because of the apparent time nodes from NGS and GS, it was practically possible to identify outliers by including the interannual and the seasonal dynamics. (2) It has high LAI values throughout the year and an LAI temporal character is like the EBF type within the year. Moreover, this kind of woody savanna was found mainly in the tropics, which could detect outliers via the $Q_{75}$ TD method. (3) It has no apparent seasonal variation characteristics, such as for samples No. 146 and No. 181, with a low LAI value from July to October every year. A previous study reported on the variations in the LAI during the wet and dry seasons. For instance, the LAI estimates ranged between 3.0 and 7.0 $m^2 \cdot m^{-2}$ during the wet season, whereas for the dry season, most parts had LAI estimates ranging between 0 and 3.5 $m^2 \cdot m^{-2}$ [64]. Nevertheless, sample No. 181 also showed a higher LAI value between July and October 2003. To validate the LAI measurements in this zone, we should increase the plot-level LAI monitoring and include high-spatiotemporal-resolution remotely sensed data in future studies [74].

## 5. Conclusions

The results obtained from this study proved that the outliers present in the MODIS LAI dataset could be identified by including temporal patterns in the post-processing. In addition, after employing the interannual dynamics, the $R^2$ between the MODIS and ground-based LAI observation improved from 0.51 to 0.54. Correspondingly, the $R^2$ between the GM LAI and MODIS LAI increased from 0.51 to 0.88 when the outliers were removed based on the seasonal dynamics. Additionally, the $R^2$ between the GM LAI and

MODIS LAI increased from 0.51 to 0.90 after eliminating the outliers in the MODIS LAI measurements detected based on the interannual and seasonal dynamics.

Moreover, we validated our results against 433 LAI measurements collected from 52 ground stations. In summary, 50, and 92, outliers of MODIS LAI can be determined when defining outliers in terms of 1.5 and 1.0 times *sdd*, respectively. Notably, the seasonal patterns are critical to identifying LAI outliers in post-processing. The denoising of a growing season can, respectively, identify 33 and 51 outliers, where the proportion of identified outliers was 66.00% and 55.43%, which is superior to the interannual patterns. The original average of MODIS LAI values was 2.20 across 433 measurements, whereas this value reached to 2.44 after post-processing, indicating that the original MODIS LAI measurements were underestimated by 10%. Our proposed methodology effectively identified the outliers in the MODIS LAI datasets incorporating the interannual and the seasonal patterns; this approach has the potential to reduce the interference for other MODIS post-processing methods, such as S-G, DL, etc. The results from this study provide a new theoretical support for the MODIS LAI post-processing.

**Supplementary Materials:** The following supporting information can be downloaded at: https://www.mdpi.com/article/10.3390/rs15205042/s1, Figure S1: Vegetation phonological information. (a–c) represent the DOY of green-up for single-growth cycle vegetation and the first and second cycle of double-growth-cycle vegetation, respectively; Figure S2: Identification of outliers of sample points (form 2003 yr to 2010 yr); Table S1: The basic information of single growth-cycle sample points.; Table S2: The basic information of double-growth-cycle sample points. Table S3: the phonological information of ground measurements LAI stations; Figure S2 attachment supplement the other samples' denoising effect of sample points; Figure S3 attachment supplement the results from the other sites of Figure 5 in the manuscript.

**Author Contributions:** Conceptualization, M.X. and B.M.; methodology, B.M. and M.X.; software, B.M. and M.X.; validation, M.X. and B.M.; data collection, B.M.; writing—original draft preparation, B.M. and M.X.; formal analysis, B.M. and M.X.; writing—review and editing, M.X. and B.M.; supervision, M.X.; project administration, M.X. and B.M.; funding acquisition, M.X. All authors worked together to design this work. All authors have read and agreed to the published version of the manuscript.

**Funding:** This research was supported by Guangdong Science and Technology plan project, the construction of Jiangmen Laboratory of Carbon Science and Technology, Hong Kong University of Science and Technology (2022B1212040001) and the National Key R&D Program of China Climate Change Impact and Adaptation in Major Countries along the Belt and Road (2018YFA0606500).

**Data Availability Statement:** The LAI data presented in this study are available online: https://lpdaac.usgs.gov/products/mcd15a3hv061/, accessed on 4 September 2023 (https://lpdaac.usgs.gov/products/mcd15a3hv061/, accessed on 12 August 2023). The phenology data (MCD12Q2) presented in this study are available online: https://lpdaac.usgs.gov/products/mcd12q2v061/, accessed on 4 September 2023 (https://lpdaac.usgs.gov/products/mcd12q2v061/, accessed on 12 August 2023). In addition, the land cover type data is here: (https://lpdaac.usgs.gov/products/mcd12q1v061/, accessed on 12 August 2023).

**Acknowledgments:** Thanks to Mojolaoluwa Toluwalase Daramola and Farhan Mustafa for the revision of English grammar in the Manuscript. Thanks to the editors and reviewers for their comments.

**Conflicts of Interest:** The authors declare no conflict of interest.

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
