# Peer review of "Identifying Outliers of the MODIS Leaf Area Index Data by Including Temporal Patterns in Post-Processing"

_remotesensing, doi:10.3390/rs15205042_

Round 1

Reviewer 1 Report

No novelty in the paper is seen.

The paper is very poorly presented. Figures are not of international standards.

Paper is written well but nothing distinguishing is outcome of the paper.

Reviewer 2 Report

The manuscript, titled "Identifying Outliers of MODIS Leaf Area Index Data by Incorporating Temporal Patterns in Post-Processing," authored by Ma and Xu, addresses the issue of outliers within MODIS leaf area index (LAI) data. The authors propose a novel method to handle these outliers by integrating temporal patterns into the post-processing procedure. Their approach is evaluated through empirical experiments and comparisons with existing outlier detection methodologies. This subject is intriguing and relevant to the domain of Remote Sensing. However, certain aspects require further refinement and attention before the manuscript can be considered suitable for publication.

One notable area for improvement is the literature review, particularly within lines 59-69. It would be beneficial to enhance this section by providing a more comprehensive assessment of the advantages and shortcomings associated with the various methods discussed, as opposed to merely listing them.

Moreover, in lines 123-127, the content appears to focus more on detailing the actions undertaken by the authors rather than clearly articulating the study's objectives. Additionally, items 1 and 2 within this context seem to convey similar information, necessitating clarification and distinction.

The quality of Figure 5 requires enhancement, as the text size appears excessively small and affects the figure's overall legibility.

The results presented in your manuscript demonstrate that your proposed method, incorporating interannual and seasonal patterns, effectively identifies a greater number of outliers, as evident in Figures 6 to 8. Nonetheless, it is important to acknowledge that your method also identifies LAIs that may not truly qualify as outliers. It is recommended that you thoroughly evaluate and address this issue within your manuscript, providing a comprehensive discussion of its implications.

Furthermore, numerous typographical errors and redundant sentences are present throughout the manuscript. It is imperative to meticulously proofread the entire manuscript for such issues. As an example, on line 116, "MODOS" should be corrected to "MODIS."

Reviewer 3 Report

This is a very well-written manuscript, and addresses all the questions/concerns of the reader in detail. I have a few notes that might improve the manuscript a bit:

- discuss in a bit more detail the structural variability of the studied sites (i.e. the vegetation species, their specific physical characteristics that could affect the LAI measurement both from the satellite point of view and the ground measurement

- How does that spatial/structural variability affect your method. Does that introduce some sort of bias in specific sites? How would/have you address(ed) that?

- The discussion section I think should be more extensive, comparing the state-of-the-art methods, and how they are compared to your method.

- discuss more how this work can help future research, how can it be used to improve maybe not only the post-processing but also the data acquisition and methodology used.

Again this is a very well-written manuscript and these are just suggestions that might make it a bit more extensive and address some of the reader's questions.

The english looks good to me. Maybe some minor changes needed when proof-reading, but nothing major I think.

Reviewer 4 Report

Extensive and thorough paper. Well done. Just a few comments in the attached pdf. This work on the identification of LAI outliers is beneficial to this community.

Round 2

Reviewer 2 Report

All my questions have been addressed, and it can be accepted after small text editing to avoid mis-spellings.
